# Optimizing Performance of Feedforward and Convolutional Neural Networks through Dynamic Activation Functions

## Abstract

In recent developments within the domain of deep learning, training algorithms have led to significant breakthroughs across diverse domains including speech, text, images, and video processing. While the research around deeper network architectures, notably exemplified by ResNet's expansive 152-layer structures, has yielded remarkable outcomes, the exploration of shallow Convolutional Neural Networks (CNN) remains an area for further exploration. Activation functions, crucial in introducing non-linearity within neural networks, have driven substantial advancements. In this paper, we delve into hidden layer activations, particularly examining their complex piece-wise linear attributes. Our comprehensive experiments showcase the superior efficacy of these piece-wise linear activations over traditional Rectified Linear Units across various architectures. We propose a novel Adaptive Activation algorithm, AdAct, exhibiting promising performance improvements in diverse CNN and multilayer perceptron configurations, thereby presenting compelling results in support of it's usage.

## 1 Introduction

Convolutional Neural Networks (CNNs) are central in image-specific tasks, serving as feature extractors that eliminate the need for explicit feature engineering in image classification. Their applications ranges from diabetic retinopathy screening (1), lesion detection (2), skin lesion classification (3), human action recognition (4), face recognition (5) and document analysis (6).

Despite their widespread use, CNNs grapple with certain limitations, including poorly understood shift-invariance, data overfitting, and reliance on oversimplified nonlinear activation functions like ReLU (7) and leaky ReLU (7). Nonlinear activation functions like ReLU and leaky ReLU have gained prominence in computer vision (8) and deep neural networks (9). Though less complex than sigmoids (7) or hyperbolic tangent functions (Tanh) (10), they partially address the vanishing gradient problem (11). However, optimal results might require various activations for individual filters in an image classification CNN with multiple filters. The ideal number of filters for specific applications remains an open area for exploration.

While these activations enable universal approximation in multilayer perceptrons, endeavors to devise adaptive or fixed PLA functions [(12), (13),(14),(15)] have emerged. Notably, adaptive activation functions for deep CNNs are introduced in (16), where the author trains the curve's slope and hinges using gradient descent techniques. The promising results in CIFAR-10, CIFAR-100 image recognition datasets (17), and high-energy physics involving Higgs boson decay modes (18) underscore their performance.

The main contribution of this paper is in understanding the usage of complex piece-wise linear activations, detailed through comprehensive experiments across a spectrum of neural network architectures. The paper offers a deep exploration of these activations in contrast to conventional Rectified Linear Units (ReLUs), showcasing their superior efficacy in CNNs and multilayer perceptrons (MLP). Our proposed adaptive activation algorithm, AdAct, exhibits promising enhancements in performance across various datasets, showcasing its potential as a robust alternative to conventional fixed activation functions. Our research work significantly advances the understanding of activation functions in neural networks, opening avenues for more nuanced design choices for improved model performance across various applications.

The remainder of this paper is organized as follows: Section 2 offers an overview of the CNN structure, notations, and existing literature. In Section 3, we delve into a detailed examination of trainable piecewise linear functions and their mathematical underpinnings. Section 4 focuses on elucidating the computational complexities associated with different algorithms. It presents experimental findings across a range of approximation and classification datasets. Additionally, this section includes results from shallow CNN experiments and insights from transfer learning applied to deep CNNs using the CIFAR-10 dataset. Finally, Section 7 discusses supplementary work and draws conclusions based on the findings presented in this paper.

## 2 Prior Work

### 2.1 Structure and Notation

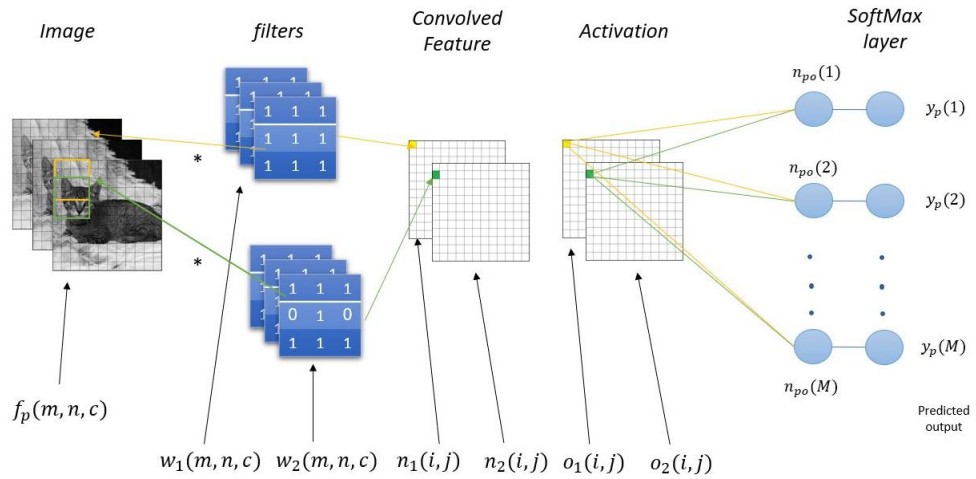

Figure 1: Shallow CNN with Linear softmax cross-entropy classifier

In Figure 1, $\mathbf{f}_p$ denote the $p^{th}$ input image and let $i_c(p)$ denote the correct class number of the $p^{th}$ pattern, where $p$ varies from 1 to $N_v$, and $N_v$ is the total number of training images or patterns. During forward propagation, a filter of size $N_f$ x $N_f$ is convolved over the image $f_1$ with $N_r$ rows $N_c$ columns. The number of channels is denoted by $C$, where color input images have $C$ equal to 3 and grayscale images have $C$ equal to 1. The net function output is $\mathbf{n}_p = \mathbf{t}_r + \sum_{c=1}^{C} \mathbf{W}_f * \mathbf{f}_p$, where, $\mathbf{n}_p$ is of size $(K \times M_o \times N_o)$, where $K$ is the number of filters, $M_o$ is the height of the convolved image output and $N_o$ is the width of the convolved image output. $\mathbf{W}_f$ is the filter of size $(K \times N_f \times N_f \times C)$. The threshold vector $\mathbf{t}_r$ is added to the net function output. The stride $s$ is the number of filter shifts over input images. Note that the output $\mathbf{n}_p$ is a threshold plus a sum of $C$ separate 2-D convolution, rather than a 3-D convolution.

To achieve non-linearity, the convolved image with element $\mathbf{n}_p$ is passed through a ReLU activation as $\mathbf{O}_p = f'(\mathbf{n}_p)$, where $\mathbf{O}_p$ is the filter's hidden unit activation output of size $(K \times N_{rb} \times N_{cb})$, where $N_{rb} \times N_{cb}$ is the row and column size of the output of the convolved image respectively. The net function $\mathbf{n}_{po}$ for $i^{th}$ element of the CNN's output layer for the $p^{th}$ pattern is $\mathbf{n}_{po} = \mathbf{t}_o + \sum_{m=1}^{M_o} \sum_{n=1}^{N_o} \sum_{k=1}^{K} \mathbf{W}_o * \mathbf{O}_p$ where $\mathbf{W_o}$ is the 4-dimensional matrix of size $(M \times M_o \times N_o \times K)$, which connects hidden unit activation outputs to the output layer net vector $\mathbf{n}_{po}$. $\mathbf{O}_p$ is a 3-dimensional hidden unit activation output matrix of size $(M_o \times N_o \times K)$ and $\mathbf{t_o}$ is the vector of biases added to net output function. Before computing the final error, the vector $\mathbf{n}_{po}$ undergoes an activation process, commonly through functions like softmax. Subsequently, the cross-entropy loss function (7) is employed to gauge the model's performance. The optimization objective involves minimizing the loss function $E_{ce}$ by adjusting the unknown weights. A common optimizer for CNN weights training is the Adam optimizer (19). It boasts computational efficiency and ease of implementation compared to the optimal learning factor (20). Leveraging momentum and adaptive learning rates, it accelerates convergence—a trait inherited from RMSProp (21) and AdaGrad (22).

## 2.2   Scaled conjugate gradient algorithm

The conjugate gradient algorithm (7) conducts line searches in a conjugate direction and exhibits faster convergence compared to the backpropagation algorithm. The scaled conjugate gradient (SCG), is a general unconstrained optimization technique to efficiently train CNN's (7). During training, a direction vector is derived from the gradient $\mathbf{g}$, where $\mathbf{p}$ is updated as $\mathbf{p} \leftarrow -\mathbf{g} + B_1 \cdot \mathbf{p}$. Here, $\mathbf{p}$ represents $\text{vec}(\mathbf{P}, \mathbf{P_{oh}}, \mathbf{P_{oi}})$, and $\mathbf{P}$, $\mathbf{P_{oi}}$, and $\mathbf{P_{oh}}$ denote the direction vectors. The ratio $B_1$ is determined from the gradient energies of two consecutive iterations, and this direction vector updates all weights simultaneously as $\mathbf{w} \leftarrow \mathbf{w} + z \cdot \mathbf{p}$. The Conjugate Gradient algorithm avoids Hessian matrix inversions so its computational cost remains at $O(\mathbf{w})$, where $\mathbf{w}$ denotes the size of the weight vector. While heuristic scaling adapts learning factor $z$ for faster convergence, Output Weight Optimization Backpropagation (OWO-BP) presents a non-heuristic Optimal Learning Factor (OLF) derived from error Taylor's series as in (21).

## 2.3   Levenberg-Marquardt algorithm

The Levenberg-Marquardt (LM) algorithm (23) merges the speed of steepest descent with Newton's method's accuracy. It introduces a damping parameter $\lambda$ to modify $\mathbf{H}_{LM} = \mathbf{H} + \lambda \cdot \mathbf{I}$, addressing potential issues with the Hessian matrix and ensuring nonsingularity. $\lambda$ balances first and second-order effects: a small value leans toward Newton's method, while a larger one resembles steepest descent. However, LM is better suited for smaller datasets due to scalability limitations (23). (21) discusses LM in detail.

## 2.4   Basic MOLF

In the fundamental MOLF based MLP training process, the input weight matrix, denoted as $\mathbf{W}$, is initialized randomly using zero-mean Gaussian random numbers. Initializing the output weight matrix, $\mathbf{W}_\text{o}$, involves employing output weight optimization (OWO) techniques (24). OWO minimizes the Mean Squared Error (MSE) function with regard to $\mathbf{W}_\text{o}$ by solving a system of $M$ sets of $N_u$ equations in $N_u$ unknowns, defined by

$$\mathbf{C} = \mathbf{R} \cdot \mathbf{W_o^T} \tag{1}$$

The cross-correlation matrix, $\mathbf{C}$, and the auto-correlation matrix, $\mathbf{R}$, are respectively represented as $\mathbf{C} = \frac{1}{N_v} \sum_{p=1}^{N_v} \mathbf{X}_\text{ap} \cdot \mathbf{t}_\text{p}^T$ and $\mathbf{R} = \frac{1}{N_v} \sum_{p=1}^{N_v} \mathbf{X}_\text{ap} \cdot \mathbf{X}_\text{ap}^T$. In MOLF, the pivotal strategy involves utilizing an $N_h$-dimensional learning factor vector, $\mathbf{z}$, and solving the equation as :

$$\mathbf{H}_\text{molf} \cdot \mathbf{z} = \mathbf{g}_\text{molf} \tag{2}$$

where $\mathbf{H}_\text{molf}$ and $\mathbf{g}_\text{molf}$ represent the Hessian and negative gradient, respectively, concerning the error and $\mathbf{z}$. Detailed algorithmic information is available in (21).

## 3   Proposed Work

Piecewise linear functions utilize ReLU activations as their primary components (9). Activations like sigmoid and Tanh can be effectively approximated using ReLU units. Numerous studies have delved into adaptive PLA functions within MLPs and deep learning contexts (13; 16). One breakthrough includes hybrid piecewise linear units (PLU) that combines Tanh and ReLU activations into a single function (12). Figure 2 illustrates how PLU combines these activations. (12) concludes that fixed PLUs outperform ReLU functions due to their representation using a greater number of hinges. However, the fixed PLA function comprises only three linear segments (with hinges $H = 3$) and remains non-adaptive until the $\alpha$ parameter is trained in each iteration. As $H$ remains fixed without significant training, it lacks the capacity for universal approximation (25). An alternative, known as the piecewise linear activation (PLA), has been demonstrated in (16), specifically designed for deep networks to accommodate trainable PLAs. This method significantly outperforms fixed PLAs, enabling the generation of more complex curves. The adaptive nature of these activations allows for more intricate curve representations.In the cited study (16), an adaptive PLA unit is introduced, where the number of hinges $H$ is a user-chosen hyperparameter. Optimal results were observed for CIFAR-10 data

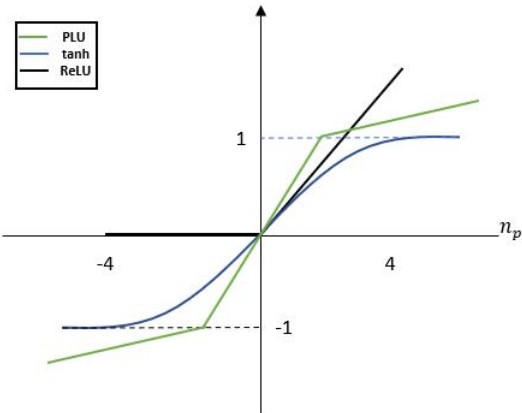

Figure 2: Fixed Piecewise Linear activations

using $H = 5$ and $H = 2$, and for CIFAR-100 data using $H = 2$ and $H = 1$ (no activation hinge training). However, the initialization method for these adaptive activations remains unspecified. The equation defining the adaptive activation function is given as: $\mathbf{o}_p = \max(0, \mathbf{n}_p) + \sum_{s=1}^{H} \mathbf{a}^s \cdot \max(0, -\mathbf{n}_p + \mathbf{b}^s)$, where, $\mathbf{a}^s$ and $\mathbf{b}^s$ are learned using gradient descent, with $\mathbf{a}^s$ controlling the slopes of linear segments and $\mathbf{b}^s$ determining sample point locations. Figure 3 demonstrates an adaptive piecewise linear function with slope $a$ as 0.2 and $b$ as 0, while Figure 4 shows a similar function with slope $a$ as -0.2 and $b$ as -0.5.

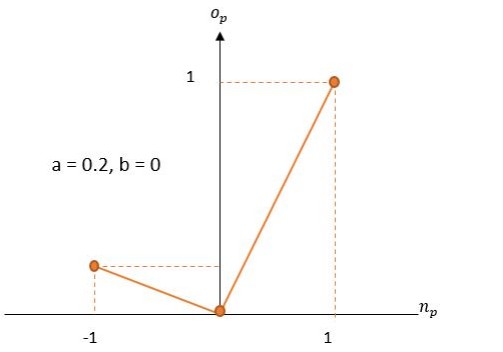
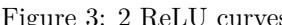

Figure 3: 2 ReLU curves

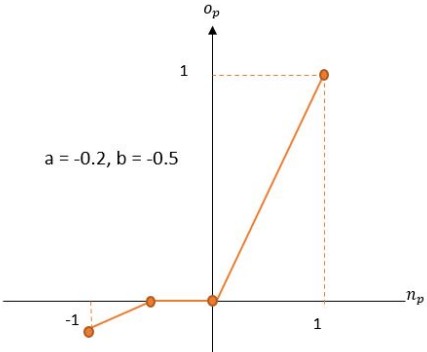

Figure 4: 4 ReLU curves

### 3.1 Mathematical Background

Adaptive PLA dynamically adjusts the positions and gradients of the hinge points. A network featuring only two hinge points surpasses ReLU activation for specific applications as discussed in the experimental section. Notably, a single hinge suffices on the piecewise linear curve to approximate a linear output, while a more substantial number of hinge sets is necessary to approximate a quadratic output. Hence, for complex datasets, an adequate number of hinges becomes imperative, avoiding the need for additional hidden layers and filters, thereby reducing training times. Our study extends to the application of PLA in CNN's (26). Initial findings highlight the capability of PLA to emulate various existing activation functions. For instance, the net function $n_1$ of a CNN filter is $\mathbf{n}_1 = t + \mathbf{w}_i \cdot \mathbf{x}$, where $t$ is the threshold, $\mathbf{w}_i$ is the filter weight, and $\mathbf{x}$ is the input to the net function. The filter can be represented as $\{\mathbf{w}_i, t\}$. The continuous PLA $f(n_1)$ is

$$f(n_1) = \sum_{k=1}^{N_s} a_k \cdot r(n_1 - ns_k) \tag{3}$$

where $N_s$ denotes the number of segments in the piecewise linear curve, $r()$ denotes a ramp (ReLU) activation, and $ns_k$ is the net function value at which the $k^{th}$ ramp switches on. Figures 5 and 6 show approximate sigmoid curves generated using ReLU activations where figure 5 has $N_s = 2$ ReLU curves and figure 6 has $N_s = 4$ ReLU curves. Comparing the two figures, we see that larger values of $N_s$ lead to better approximation. The contribution of $f(n_1)$ to the $j^{th}$ net function $n_2(j)$ in the following layer is $+n_2(j) = f(n_1) \cdot w_o(j)$. Decomposing the PLA into its $N_s$ components, we can write $+n_2(j) = \sum_{k=1}^{N_s} a_k \cdot r(n_1 - ns_k) \cdot w_o(j) = \sum_{k=1}^{N_s} w_o'(j,k) \cdot r(n_1(k))$, where $w_o'(j,k)$ is $a_k \cdot w_o(j)$ and $n_1(k)$ is $n_1 - ns_k$. A single PLA for filter $\{\mathbf{w_i}, t\}$ has now become $N_s$ ReLU activations $f(n_1(k))$ for $N_s$ filters, where each ramp $r(n_1 - d_k)$, is the activation output of a filter. These $N_s$ filters are identical except for their thresholds. Although ReLU activations are efficiently computed, they have the disadvantage that back-propagating through the network activates a ReLU unit only when the net values are positive and zero; this leads to problems such as dead neurons(27), which means if a neuron is not activated initially or during training, it is deactivated. This means it will never turn on, causing gradients to be zero, leading to no training of weights. Such ReLU units are called dying ReLU(28).

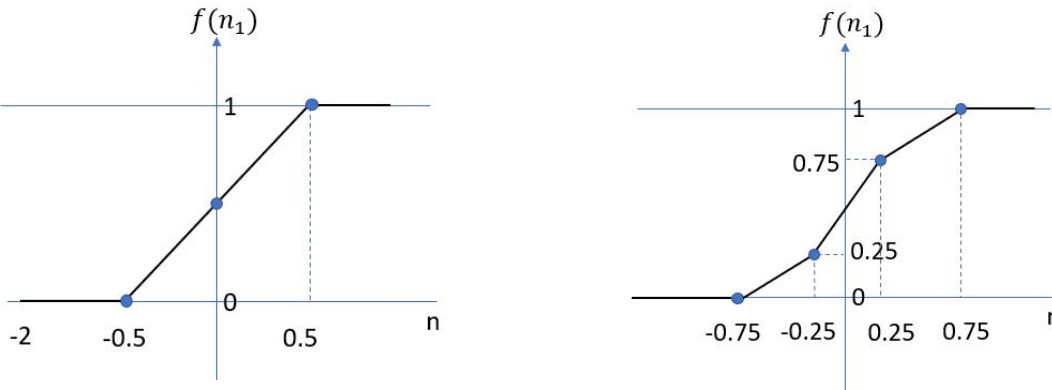

Figure 5: Approximate sigmoid using 2 ReLU curves

Figure 6: Approximate sigmoid using 4 ReLU curves

### 3.2 Piecewise Linear Activations

Existing PLAs, as discussed, have their limitations, notably when encountering minimal distances between the heights of two hinges during training. In the context of the multiple ramps function, each term within the sum encompasses a ramp function accompanied by a coefficient. When the difference between the value of $n_1$ and its corresponding index is non-positive, the ramp function yields 0, thereby nullifying the contribution of that term to the overall sum. Consequently, there are instances where the function values may equate to zero. We define a highly robust PLA capable of initialization through diverse pre-defined activations like ReLU (7) and leaky ReLU (7), enabling differentiability.

Figure 7 illustrates a PLA for K hidden units, comprising multiple ramps defined by $ns(1,k)$ as the initial hinge of the $k^{th}$ hidden unit and $a(1,k)$ as its activation value. These hinge values $ns$ remain constant throughout training. The depicted piecewise linear curve passes through the activations of each of the 7 hinges. We denote the total number of hinges as $H$. Let $s$ represent the maximum value of a net function, and $r$ denote its minimum value. The activations are computed between two points; for instance, net values between the first two hinges utilize $ns(1,k)$ as $m_1$ and $ns(2,k)$ as $m_2$. Similarly, activations between subsequent hinges involve denoting $ns(2,k)$ as $m_1$ and $ns(3,k)$ as $m_2$. We perform this process for $H$ hinges. For each hinge, $m_1$ and $m_2$ are calculated using $m_1 = \lceil \frac{n_p}{\delta ns} \rceil$ and $m_2 = m_1 + 1$. Subsequently, with the net function $n_p(k)$ given, $o_p(k)$ is computed as follows:

$$w_{1p}(k) = \frac{ns(m_2, k) - n_p(k)}{ns(m_2, k) - ns(m_1, k)} \tag{4}$$

$$w_{2p}(k) = \frac{n_p(k) - ns(m_1, k)}{ns(m_2, k) - ns(m_1, k)} \tag{5}$$

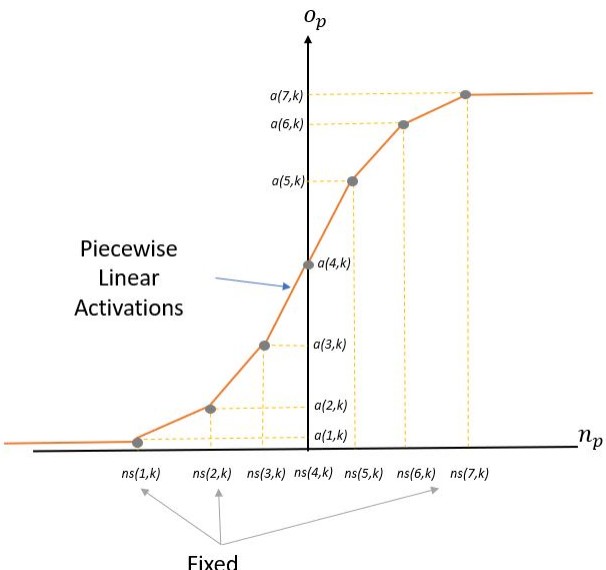

Figure 7: Piecewise Linear Curve

$$o_p(k) = \begin{Bmatrix} a(H,k) & for & n_p(k) > s \\ w_{1p}(k) \cdot a(m_1,k) + w_{2p}(k) \cdot a(m_2,k) & for & s > n_p(k) > r \\ a(1,k) & for & n_p(k) < r \end{Bmatrix} \tag{6}$$

where $w_{1p}(k)$ and $w_{2p}(k)$ represent the slope equation for each of the two hinges for the $k^{th}$ hidden unit. $n_p(k)$ denotes the $p^{th}$ pattern and $k^{th}$ hidden unit net function, while $o_p(k)$ represents its activation function. For all activation values less than $ns(1,k)$, the slope is zero, thus $n_p(k) = a(1,k)$. Similarly, for all values greater than $ns(H,k)$, the slope is zero, hence $n_p(k) = a(H,k)$. The proposed piecewise linear activations $\mathbf{A}$ are trained using the steepest descent method. The negative gradient matrix $\mathbf{G_a}$ with respect to $E_{ce}$ is calculated as follows:

$$g_a(k,m) = -\frac{\partial E_{ce}}{\partial a(k,m)} \tag{7}$$

where $k$ is the hidden unit number and $m$ is the $H^{th}$ hinge.

$$g_a(k,m) = \frac{2}{N_v} \sum_{p=1}^{N_v} \sum_{i=1}^{M} (t_p(i) - y_p(i)) \cdot \frac{\partial y_p(i)}{\partial a(u,m)} \tag{8}$$

$$\frac{\partial y_p(i)}{\partial a(u,m)} = w_{oh}(i,u) \cdot \frac{\partial o_p(i)}{\partial a(u,m)} \tag{9}$$

$$\frac{\partial o_p(i)}{\partial a(u,m)} = w_{oh}(i,u) \cdot ((\delta(m-m_1) \cdot w_1(p,u)) + (\delta(m-m_2) \cdot w_2(p,u))) \tag{10}$$

where, for the $p^{th}$ pattern and $k^{th}$ hidden unit's net value, we determine $m_1$ and $m_2$ based on the $p^{th}$ pattern of the $k^{th}$ hidden unit's net value falling between the fixed piecewise linear hinge values $m_1$ and $m_2$ of the $u^{th}$ hidden unit, as detailed in the search algorithm provided in the appendix. Subsequently, we obtain $w_1(p,u)$ and $w_2(p,u)$ from equations 4 and 5. Utilizing a search algorithm described in (26), we locate the correct

hinge $m$ for a specific pattern's hidden unit. Equation 10 resolves the $p^{th}$ pattern's $u^{th}$ hidden unit of the PLAs, accumulating the gradient for all $p^{th}$ patterns of their respective $u^{th}$ hidden units. Adam optimizer (19) is used to determine the learning factor and update the activation weights. The weight updates are performed as follows:

$$\mathbf{A} = \mathbf{A} + z \cdot \mathbf{G_a} \tag{11}$$

Using the gradient $\mathbf{G_a}$, the optimal learning factor for activations training is calculated as, The activation function vector $\mathbf{o}_p$ can be related to its gradient as,

$$\mathbf{o}_p(k) = w_1(p,k) \cdot [a(k,m_1) + z \cdot g_o(k,m_1)] + w_2(p,k) \cdot [a(k,m_2) + z \cdot g_o(k,m_2)] \tag{12}$$

The first partial derivative of E with respect to z is

$$\frac{\partial E}{\partial z} = \frac{2}{N_v} \sum_{p=1}^{N_v} \sum_{i=1}^{M} (t_p(i) - y_p(i)) \cdot \frac{\partial y_p(i)}{\partial z} \tag{13}$$

where

$$\frac{\partial y_p(i)}{\partial z} = \sum_{k=1}^{N_h} w_{oh}(i,k) \cdot ((w_1(p,k) \cdot g_o(k,m_1)) + (w_2(p,k) \cdot g_o(k,m_2))) \tag{14}$$

where, $m_1$ and $m_2$ for the $p^{th}$ pattern and $k^{th}$ hidden unit of the net vector $n_p(k)$ is again found, and find $g_o(k,m_1)$ and $g_o(k,m_2)$ from the gradient calculated from equations (8, 9, 10). Also, the Gauss-Newton (29) approximation of the second partial is

$$\frac{\partial^2 E(z)}{\partial z^2} = \frac{2}{N_v} \sum_{p=1}^{N_v} \sum_{i=1}^{M} \left[ \frac{\partial y_p(i)}{\partial z} \right]^2 \tag{15}$$

Thus, the learning factor is calculated as

$$z = \frac{\frac{-\partial^2 E(z)}{\partial z^2}}{\frac{\partial E}{\partial z}} \tag{16}$$

After finding the optimal learning factor, the PLAs, $\mathbf{A}$, are updated in a given iteration as

$$\mathbf{A} = \mathbf{A} + z \cdot \mathbf{G_a} \tag{17}$$

where $\mathbf{z}$ is a scalar optimal learning factor and $\mathbf{G_a}$ is the gradient matrix calculated in equation 16 and 7. A pseudo-code for the proposed AdAct algorithm is as follows:

---
**Algorithm 1** AdAct algorithm
---
1: Initialize $\mathbf{W}, \mathbf{W_{oi}}, \mathbf{W_{oh}}, N_{it}$
2: Initialize Fixed hinges **ns** and hinge activation **a** , it$\leftarrow 0$
3: **while** it $< N_{it}$ **do**
4:     Find gradient $\mathbf{G}_a$ and $g_{molf}$ from equations 7 and 2 and solve for $z$ using OLS.
5:     Calculate gradient and learning factor for activation from equation 7 and equation 16 respectively and update the activations as in equation 17.
6:     **OWO step** : Solve equation (1) to obtain $\mathbf{W_o}$
7:     it $\leftarrow$ it $+ 1$
8: **end while**
---

## 4 Experimental Results

We present the experimental results demonstrating the relative performance of our proposed algorithm across diverse approximation and classification datasets. Moreover, we delve into the comparison of network performance among various methodologies, including AdAct (30), MOLF (31), CG-MLP (21), Scaled conjugate gradient (SCG) (21) and LM (23) across approximation and classification datasets. Additionally, we present results for shallow convolutional neural networks utilizing both custom architectures and deep CNNs employing transfer learning strategies.

Additional results are available in the appendix (supplementary material), which include approximations of a simple sinusoidal function and a more complex Rosenbrock function. These results yield insights into the behaviour of different activation functions. Specifically, the proposed AdAct showcase consistent outputs for the simple sinusoidal function regardless of the initial activation. Conversely, models utilizing ReLU and Leaky ReLU activations can only accurately approximate the sinusoidal function if more hidden units are incorporated. Augmenting the number of piecewise hinges, enhances the accuracy of the sinusoidal approximation but amplifies computational costs. The computational expense associated with adaptive activation encompasses the aggregate of trainable parameters and the product of hidden units and hinge counts.

### 4.1 Computational Burden

The computational burden is used to measure the time complexity for each algorithm. It indicates number of multipliers that a particular algorithm needs to process per iteration using inputs, hidden units and outputs. We calculate computational burden for the proposed AdAct algorithm along with the comparing algorithms. AdAct algorithm has number of hinges as $N_{hinges}$. Updating input weights using Newton's method or LM, requires a Hessian with $N_w$ rows and columns, whereas the Hessian used in the proposed Adapt-MOLF has only $N_h$ rows and columns. The total number of weights in the network is denoted as $N_w = M \cdot N_u + (N+1) \cdot N_h$ and $N_u = N + N_h + 1$. The number of multiplies required to solve for output weights using the OLS is given by $M_{ols} = N_u(N_u + 1)[M + \frac{1}{6}N_u(2N_u + 1) + \frac{3}{2}]$. Therefore, the total number of multiplications per training iteration for LM, SCG, CG, MOLF and AdAct algorithm is given as :

$$M_{lm} = [MN_u + 2N_h(N+1) + M(N + 6N_h + 4) + MN_u(N_u + 3N_h(N+1)) + 4N_h^4(N+1)^2] + N_w^3 + N_w^2 \quad (18)$$

$$M_{cg} = M_{scg} = [MN_u + M(N + 6N_h + 4) + MN_u(N_u + 3N_h(N+1)) + 4N_h^4(N+1)^2] + N_w^3 + N_w^2 \quad (19)$$

$$M_{molf} = M_{ols} + N_v N_h[2M + N + 2 + \frac{M(N_h + 1)}{2}] \quad (20)$$

$$M_{AdAct} = M_{molf} + N_h * N_{hinges} \quad (21)$$

Note that $M_{AdAct}$ consists of $M_{molf}$ plus the required multiplies for the number of hinges for each of the hidden unots. The computational cost of Adapt-MOLF algorithm also varies between computational cost of OWO-MOLF and OWO-Newton.

### 4.2 Approximation Datasets Results

We using the following datasets for model evaluation Oh7(32), White wine (33), twod ,Superconductivity dataset (34),F24 data (35), concerete (36), weather dataset (37). From Table 1, we can observe that Adapt-OLF is the top performer in 5 out of the 7 datasets in terms of testing MSE. The following best-performing algorithm is MOLF-Adapt for weather data with a smaller margin and is also tied in comparison to testing

| Dataset | SCG/Nh | CG-MLP/Nh | LM/Nh | MOLF/Nh | AdAct/Nh 3-hinges | AdAct/Nh 5-hinges | AdAct/Nh 9-hinges |
|---|---|---|---|---|---|---|---|
| Oh7 | 1.971/30 | 1.52/100 | **1.41**/30 | 1.51/15 | 1.49/20 | 1.46/20 | 1.44/15 |
| White Wine | 0.6/20 | 0.56/100 | 0.57/30 | 0.55/30 | **0.54/100** | 0.56/20 | **0.54/100** |
| twod | 0.5/30 | 0.23/100 | 0.17/15 | **0.149/15** | **0.149/15** | 0.18/15 | 0.15/15 |
| Super-conductor | 230.91/15 | 180.21/100 | 170.2/100 | 144.46/100 | 142.53/100 | **139.62/100** | 142.53/100 |
| F24 | 1.14/20 | 0.31/100 | 0.30/30 | 0.281/100 | 0.282/100 | 0.283/100 | **0.279/100** |
| Concrete | 61.11/5 | 34.64/30 | 32.12/20 | 32.29/100 | **30.70/100** | 34.34/10 | 35.56/100 |
| Weather | 316.68/15 | 283.23/30 | 286.27/30 | **283.20/10** | 284.39/15 | 284.34/15 | 284.73/15 |

Table 1: Comparison of 10-fold cross-validation mean square error (MSE) testing results for various approximation datasets using different models and configurations. The best-performing MSE results are highlighted in bold. MOLF is using a $Leaky - ReLU$ as activation function.

MSE for twod datasets. Adapt-OLF has slightly more parameters depending on the number of hinges, but the testing MSE is substantially reduced. The next best performer is LM for the Oh7 dataset. However, LM being a second-order method, its performance comes at a significant cost of computation – almost two orders of magnitude greater than the rest of the models proposed. The table displays the 10-fold cross-validation mean square error (MSE) testing results for various datasets across different models, each denoted by a specific configuration. The values indicate the MSE achieved by different algorithms concerning the number of hidden units (Nh) and hinges used in adaptive activation functions. The models with the best testing MSE are highlighted in bold. Observations reveal that the performance of these algorithms varies across datasets, indicating no single superior algorithm across all scenarios. For instance, for the Oh7 dataset, the LM/Nh configuration performs best, while for White Wine and twod datasets, MOLF-AdAct configurations with different hinge counts deliver the lowest MSE. Notably, more complex datasets, like F24, demonstrate a pattern where a higher number of hinges yields better performance. This suggests that the optimal choice of algorithm and its configuration heavily depends on the dataset complexity and characteristics, emphasizing the need for adaptability in selecting the suitable model for diverse data scenarios.

## 4.3 Classifier Datasets Results

| Dataset | SCG/Nh | CG-MLP/Nh | LM/Nh | MOLF/Nh | AdAct/Nh 3-hinges | AdAct/Nh 5-hinges | AdAct/Nh 9-hinges |
|---|---|---|---|---|---|---|---|
| GongTrn | 10.28/100 | 10.46/100 | 8.94/30 | **8.62**/30 | 8.65/30 | 8.64/30 | 8.72/30 |
| Comf18 | 15.69/100 | 14.50/100 | 12.63/5 | 11.83/20 | 11.93/30 | 11.87/30 | **11.79/30** |
| f17c | 3.22/100 | 3.69/100 | 3.96/100 | 2.45/100 | 2.38/100 | 2.41/100 | **2.34/100** |
| Speechless | 44.26/100 | 43.07/100 | 39.72/100 | 36.65/100 | **35.96/100** | 38.94/100 | 37.6/100 |
| Cover | 27.39/100 | 29.87/100 | NA | 20.1/30 | 19.43/30 | 19.47/30 | **19.42/30** |
| Scrap | 25.58/100 | 20.77/100 | NA | 19.9/100 | 19.57/100 | **18.8/100** | 19.2/100 |

Table 2: 10-fold cross validation Percentage of Error (PE) testing results for classification dataset, (best testing MSE is in bold)

For Classification tasks, we utilized various datasets for evaluating our models, including Gongtrn data (38), Comf18 data (39), Speechless data (39), Cover data (40), Scrap data (39), and f17c (39). Analyzing Table 2, it's evident that Adapt-OLF emerges as the top performer in 5 out of 6 datasets based on testing MSE. MOLF-Adapt for weather data also exhibits competitive performance, albeit with a smaller margin. Notably, LM, despite its computational cost, isn't suitable for pixel-based inputs, limiting its applicability in numerous image classification scenarios.

### 4.4 CNN Results

### 4.4.1 Shallow CNN results

We conducted experiments using shallow CNNs with ReLU, leaky ReLU, and adaptive activations with one, two, and three VGG blocks (41) on the CIFAR-10 dataset for benchmarking.

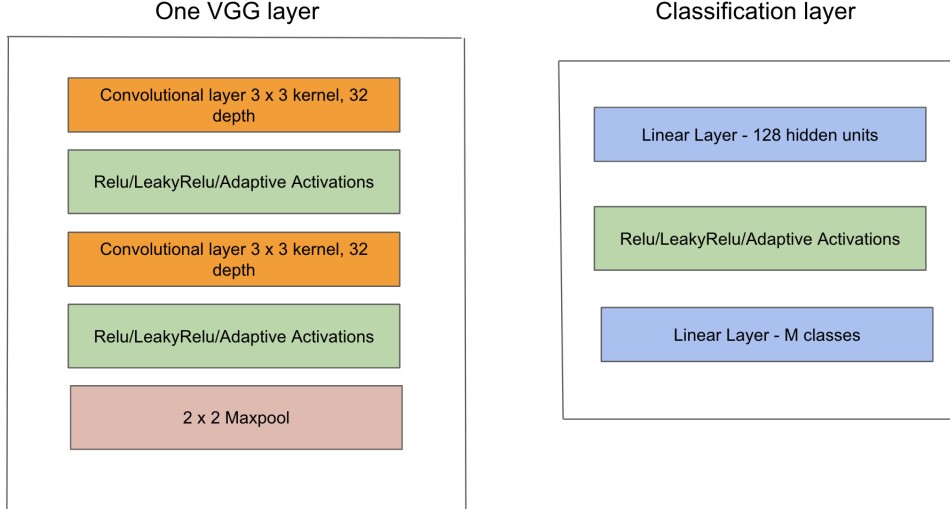

Figure 8: 1 VGG layer with classifier,it consists of 2 convolution layers with 3x3 kernel and 32 filters in the first and and 3rd layers. Activations are in the 2nd and 4th layers. The final layer is a 2 x 2 maxpool layer

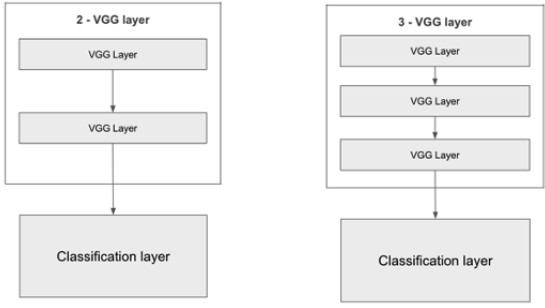

Figure 9: Two and Three VGG layer with classifier

Figure 8 illustrates the configuration of the One VGG block, comprising 2 convolution layers with 3x3 kernels, 32 filters in the first and third layers, activation in the second and fourth layers, and a final 2x2 maxpool layer. Similarly, the 2 VGG layer network includes two VGG layers, with the first layer's configuration matching the one depicted in Figure 8 and the second VGG layer consisting of convolution layers with 3x3 kernels and a depth of 64 for each convolution layer, shown in Figure 9. In the 3 VGG layer configuration, we maintain the setup of the 2 VGG layers described earlier, with an additional 3rd VGG layer following the same structure as the 2nd VGG layer but utilizing 64 filters for each convolution layer, detailed in Figure 9.

An important observation is that while the model uses ReLU activations exclusively for ReLU, and similarly for leaky ReLU, in adaptive activations, only the last VGG layer's activations are trained, leaving the rest as either ReLU or leaky ReLU. For instance, in the 2-VGG layer setup, the 2nd VGG layer's adaptive activation functions are trained, while the first VGG layer's adaptive activations are not treated as trainable

parameters. The activation range, denoted by 'n' and set at 3 samples [minimum value, 0, maximum value], represents the range of output values before the adaptive activation. In practical terms, when training a 2-VGG layer model, activations in the VGG1 layer remain untrained. The output from VGG1, the 2x2 maxpool output, serves as the input to the 2nd VGG layer, where the maximum and minimum values of the first convolution layer's output in the 2nd VGG layer are utilized. Leaky ReLU is the activation function used for non-trainable adaptive activations, and the initialization of these activations also employs leaky ReLU. The choice of activation type for initialization was based on results showcased in Table 3 for the model featuring leaky ReLU activations.Table 3 shows results for 1-VGG layers, 2-VGG layers, 3-VGG layers model on CIFAR-10(17) dataset with Glorot normal initialization From the table we can observe that as the number of VGG layer increase, adaptive activation gives better accuracy. One thing to note is that in 3-VGG layer model, only the 3rd VGG layer activations are trained, and we can observe a significant difference in the accuracy.

| Models | Adaptive | ReLU | LeakyReLU |
|---|---|---|---|
| 1 - VGG layer | **67.8** | 66.56 | 66.45 |
| 2 - VGG layers | **74.2** | 71.82 | 73.09 |
| 3 - VGG layers | **75.53** | 72.58 | 73.3 |

Table 3: 10-fold cross validation accuracy testing results on various activation functions for CIFAR-10 dataset using Glorot as weight-initialization(best testing accuracy is in bold)

### 4.4.2   Transfer Learning using Deep CNN results

In this section, we explore the utilization of two widely-used pretrained deep learning models, VGG11 and ResNet18, initially trained on the ImageNet dataset. Employing transfer learning techniques, we adapt these models for use with the CIFAR-10 dataset. This adaptation involves replacing the final classification layer with a new linear layer containing ten output classes, aligning with CIFAR-10's class count. The subsequent fine-tuning process involves a minimum of 100 iterations. Transfer learning capitalizes on the knowledge these pretrained models gained from ImageNet, allowing us to achieve commendable results with reduced data and iterations. Additionally, in models featuring adaptive activations, we implement changes to the final layers. For instance, in the ResNet18 architecture, we substitute ReLU activation functions in the last layer's basic blocks with adaptive activations. In VGG11, modifications target the activations following the 7th and 8th convolution layers, which also serve as the last two activations in the feature layer, introducing adaptive activations. Adopting adaptive activations in the final feature layer holds two primary benefits: parameter reduction and improved modeling of deeper layer's complex and abstract features (42). The results showcased in Table 4 indicate that while adaptive activations lead to better results, there is a slight increase in parameters and training time.

| Models | Adaptive Activations | ReLU Activations |
|---|---|---|
| VGG11 | **91.78** | 91.44 |
| ResNet18 | **95.30** | 95.1 |

Table 4: 10-fold cross validation testing accuracy results for classification dataset (best testing accuracy is in bold)

## 5   Conclusion and Future Work

The present study focuses on the significance of activation functions within neural networks and the promising role of AdAct algorithm, an adaptive piecewise linear activation algorithm, in comparison to traditional Rectified Linear Units (ReLUs). The exploration of both deep and shallow CNN architectures emphasizes on the efficacy of adaptive activations, demonstrating superior performance across various datasets and model complexities. Based on the experimental results, the AdAct algorithm is robust and has the ability to approximate complex functions, surpassing the limitations of fixed activation functions. While AdAct entail

increased computational demands, the attained performance levels substantiate their value, especially in scenarios requiring accurate approximation of curved outputs. Our experimentation underscores the limitations of fixed activations in capturing arbitrary functions and highlights the adaptability and convergence advantages offered by adaptive activations.

The current study adds a valuable perspective to the ongoing discourse on neural network design, urging researchers and practitioners to consider the nuanced details of activation functions for optimal performance across diverse applications. Utilizing fixed activations to approximate complex curves like sinusoids proves challenging, particularly with activation functions like ReLU and leaky ReLUs, which lack a curve in their function. However, employing adaptive activations enables the model to converge to the desired output curve, irrespective of the initial activation, necessitating a greater number of samples for achieving a smoother curve. These findings underscore the potential of AdAct algorithm and their implications in neural network design, encouraging deeper exploration into their usage and optimization for varied applications.

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

## A    Training weights by orthogonal least squares

OLS is used to solve for the output weights, pruning of hidden units (43), input units (43) and deciding on the number of hidden units in a deep learner (44). OLS is a transformation of the set of basis vectors into a set of orthogonal basis vectors thereby measuring the individual contribution to the desired output energy from each basis vector.

In an autoencoder, we are mapping from an (N+1) dimensional augmented input vector to it's reconstruction in the output layer. The output weight matrix $\mathbf{W_{oh}} \in \Re^{N \times N_h}$ and $y_p$ in elements wise will be given as

$$y_p(i) = \sum_{n=1}^{N+1} w_{oh}(i,n) \cdot x_p(n) \tag{22}$$

To solve for the output weights by regression , we minimize the MSE. In order to achieve a superior numerical computation, we define the elements of auto correlation $\mathbf{R} \in \Re^{N_h \times N_h}$ and cross correlation matrix $\mathbf{C} \in \Re^{N_h \times M}$ as follows :

$$r(n,l) = \frac{1}{N_v} \sum_{p=1}^{N_v} O_p(n) \cdot O_p(l) \qquad c(n,i) = \frac{1}{N_v} \sum_{p=1}^{N_v} O_p(n) \cdot t_p(i) \tag{23}$$

Substituting the value of $y_p(i)$ in MSE we get,

$$E = \frac{1}{N_v} \sum_{p=1}^{N_v} \sum_{m=1}^{M} [t_p(m) - \sum_{k=1}^{N_h} w_oh(i,k) \cdot O_p(k)]^2 \tag{24}$$

Differentiating with respect to $\mathbf{W_{oh}}$ and using equation 23 we get

$$\frac{\partial E}{w_{oh}(m,l)} = -2[c(l,m) - \sum_{k=1}^{N_h+1} w_{oh}(m,k)r(k,l)] \tag{25}$$

Equating equation 25 to zero we obtain a $M$ set of $N_h + 1$ linear equations in $N_h + 1$ variables. In a compact form it can be written as

$$\mathbf{R} \cdot \mathbf{W}^T = \mathbf{C} \tag{26}$$

By using orthogonal least square, the solution for computation of weights in equation 26 will speed up. For convineance, let $N_u = N_h + 1$ and the basis functions be the hidden units output $\mathbf{O} \in \Re^{(N_h+1) \times 1}$ augmented with a bias of $\mathbf{1}$. For an unordered basis function $\mathbf{O}$ of dimension $N_u$ , the $m^{th}$ orthonormal basis function $\mathbf{O}'$ is defines as « add reference »

$$O'_m = \sum_{k=1}^{m} a_{mk} \cdot O_k \tag{27}$$

Here $a_{mk}$ are the elements of triangular matrix $\mathbf{A} \in \Re^{N_u \times N_u}$

For $m = 1$

$$O'_1 = a_{11} \cdot O_1 \qquad a_{11} = \frac{1}{\|O\|} = \frac{1}{r(1,1)} \tag{28}$$

for $2 \leq m \leq N_u$, we first obtain

$$c_i = \sum_{q=1}^{i} a_{iq} \cdot r(q, m) \tag{29}$$

for $1 \leq i \leq m - 1$. Second, we set $b_m = 1$ and get

$$b_{jk} = -\sum_{i=k}^{m=1} c_i \cdot a_{ik} \tag{30}$$

for $1 \leq k \leq m - 1$. Lastly we get the coeffeicent $A_{mk}$ for the triangular matrix $\mathbf{A}$ as

$$a_{mk} = \frac{b_k}{[r(m, m) - \sum_{i=1}^{m-1} c_i^2]^2} \tag{31}$$

Once we have the orthonormal basis functions, the linear mapping weights in the orthonormal system can be found as

$$w'(i, m) = \sum_{k=1}^{m} a_{mk} c(i, k) \tag{32}$$

The orthonormal system's weights $\mathbf{W}'$ can be mapped back to the original system's weights $\mathbf{W}$ as

$$w(i, k) = \sum_{m=k}^{N_u} a_{mk} \cdot w'_o(i, m) \tag{33}$$

In an orthonormal system, the total training error can be written from MSE as

$$E = \sum_{i=1}^{M} \sum_{p=1}^{N_v} [\langle t_p(i), t_p(i) \rangle - \sum_{k=1}^{N_u} (w'(i, k))^2] \tag{34}$$

Orthogonal least square is equivalent of using the $\mathbf{QR}$ decomposition (45) and is useful when equation equation 26 is ill-conditioned meaning that the determinant of $\mathbf{R}$ is $\mathbf{0}$.

## B  Example of piecewise linear activation

To initialize PWL activation, we begin by initializing it with sigmoid(7), ReLU(7) and leaky ReLU(7) activation functions. Subsequently, we determine the total number, $H$, of units in the network. This approach can be applied individually to each hidden unit. In this study, an equal number of hinges, denoted as $n_s$, are utilized for each hidden unit. The process involves identifying the minimum and maximum hinge values derived from the network function output. To achieve this, data samples are randomly chosen from each class, and convolution is performed. The resulting convolution output yields the minimum and maximum values. The subsequent section outlines the computation for PWL activations considering $k = 1$ hidden unit. As previously mentioned, the initial activation must be determined. In this illustration, we utilize the sigmoid activation function, as depicted in Figure 10.

The figure 10 illustrates the net function, $n_p$, along the x-axis and its corresponding sigmoid activation on the y-axis, with the sigmoid's range spanning from 0 to 1. The second step involves determining the

minimum and maximum values from the convolution output, which, in this case, are selected as -4 and 4, respectively. Subsequently, a user-defined step involves selecting $n_s$ samples on the sigmoid curve. For this specific example, we opt for $H = 7$. These selected hinges and their corresponding activations are presented in Table 5. This table displays a total of $H = 7$ hinges, ranging from -4 to 4, derived from the minimum and maximum values of the net function, with their associated sigmoid activations denoted as 'a'. The final step involves plotting these chosen points onto the sigmoid curve illustrated in Figure 10. The resulting curve, after incorporating these specified points, should resemble the representation in Figure 11.

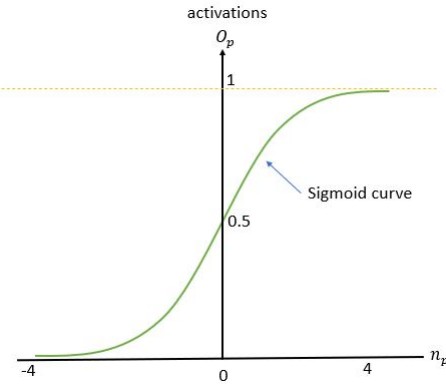

Figure 10: Sigmoid Curve

| H | 1 | 2 | 3 | 4 | 5 | 6 | 7 |
|---|---|---|---|---|---|---|---|
| Fixed hinges ($ns_1$) | -4 | -2.67 | -1. | 0 | 1. | 2.67 | 4 |
| Activations for hinges($a_1$) | 0.02 | 0.07 | 0.21 | 0.5 | 0.79 | 0.94 | 0.98 |

Table 5: PWL samples and activations for one hidden unit

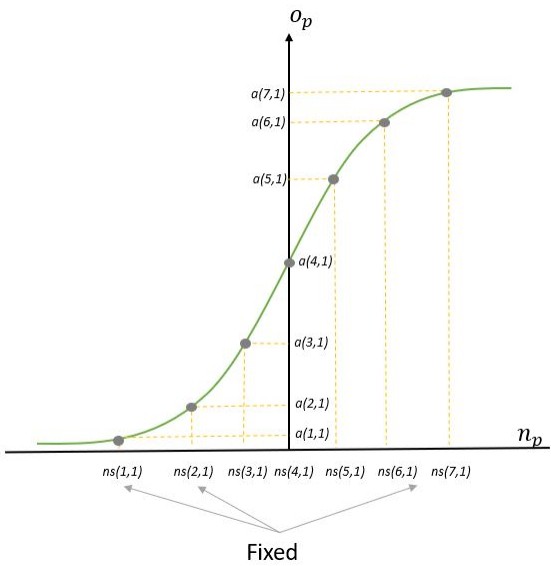

Figure 11: Sigmoid with Fixed Samples

Figure 11 is the plot for a fixed piecewise sigmoid activation for net versus activations values where 7 hinges are plotted onto the sigmoid curve. For the final piecewise linear curve, we remove the sigmoid curve and linearly join 2 points using the linear interpolation technique.

Linear interpolation involves estimating a new value of a function between two known fixed points (7).

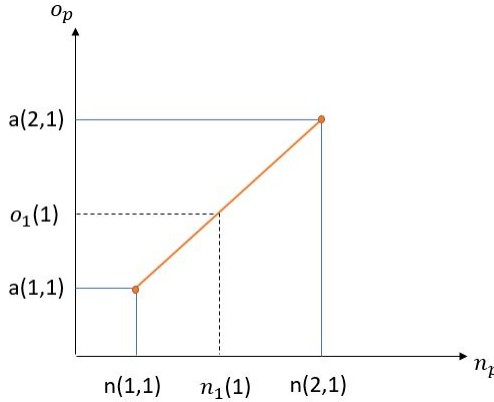

Figure 12: Linear interpolation between 2 points

Figure 12 demonstrates the application of linear interpolation between two fixed $n_s$ points. Suppose we have a new sample net value, $n_1(1)$; its corresponding activation value is depicted in the figure. To determine $o_1(1)$, the interpolation between $a(1,1)$ and $a(2,1)$ is calculated using the following equation:

$$o_1(1) = \frac{ns(2,1) - n_1(1,1)}{ns(2,1) - ns(1,1)} \cdot a(1,1) + \frac{n_1(1) - ns(1,1)}{ns(2,1) - ns(1,1)} \cdot a(2,1) \tag{35}$$

Subsequently, equation 6 is employed to compute all activation outputs. The resulting plot of net versus activation is presented in Figure 7.

## C   Adavantages of piecewise linear activation

Now we demonstrate the advantage of adaptive activations using a simple sine data function and a more complicated Rosenbrock function(46). For each of the experiments, we will compare the approximation results for MOLF algorithm explained in section 2.4 with constant activation functions such as sigmoid, Tanh,ReLU and leaky ReLU and with AdAct algorithm described in 1 with initial activations as sigmoid, Tanh, ReLU and leaky ReLU respectively.

### C.1   Sinusoidal Approximation

The sine data used for training is generated using a single feature and consists of 5000 uniformly distributed random samples within the range of 0 to $4\pi$. The resulting output is calculated as the sine function applied to these uniformly distributed random samples. For testing, we randomly select 100 uniformly distributed random samples within the range of 0 to $2\pi$. Training for both the MOLF and AdAct algorithms involves 100 iterations. Each training algorithm employs 1 hidden unit, except for the AdAct algorithm, which utilizes 20 samples. Additionally, results are presented for the MOLF algorithm using 10 hidden units. In this case, 20 samples are used, as piecewise linear activations (PLAs) would involve 10 smaller ReLU-like functions. Figure 13 displays the prediction of the MOLF model with one hidden unit after 100 iterations, employing the aforementioned fixed activations. The figure illustrates that none of the fixed activations closely approximate the sine function, as labeled the 'original target' in Figure 13a. Subsequently, when increasing the hidden units to 10, Figure 13b showcases the model's prediction with ten hidden units after 100 iterations, using the fixed activations mentioned earlier. Notably, curve-based functions like Tanh and sigmoid approximate

the sine function, whereas activation functions like ReLU and leaky ReLU, which employ piecewise linear approximations, do not approximate even with more hidden units.

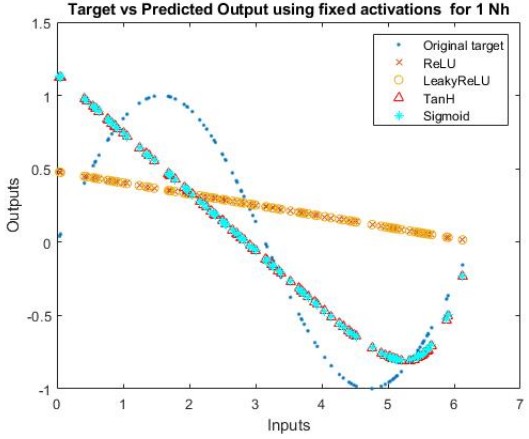

(a) MOLF with fixed activations with one hidden unit

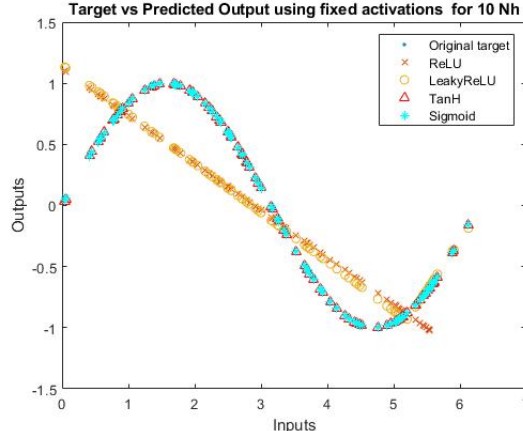

(b) MOLF with fixed activations with ten hidden units

Figure 13: MOLF with fixed activations

Figure 14 shows AdAct model prediction with ten hidden units after 100 iterations with initial activations as each of the mentioned fixed activations. From the figure 14, we can observe that the model trained using the adaptive activations approximates the function with similar output no matter what initial activations are used.

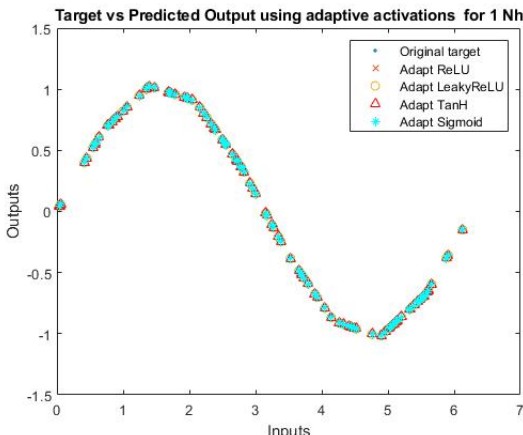

Figure 14: Adaptive activations algorithm with 20 samples

Figure 15 shows the hidden units after the model is trained for each fixed initial hidden unit. We can observe that figure 15c and figure 15d are trained with initial activation as ReLU and Leaky ReLU mimics the input versus output, which is sinusoidal which very high activation output but the activation outputs for sigmoid and Tanh are not as high as shown in figure 15a and figure 15b. We also observed that as the activation output is so high, the trained output weights were equally small, with the input weight range being close to similar to the model trained with Tanh's activation.

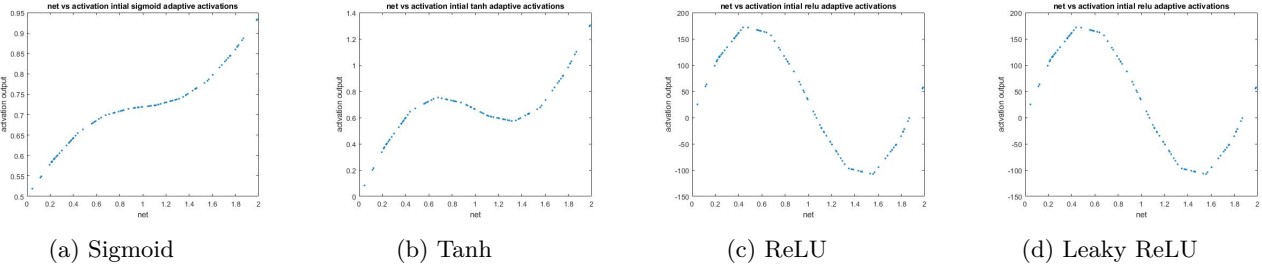

(a) Sigmoid    (b) Tanh    (c) ReLU    (d) Leaky ReLU

Figure 15: Adaptive Activation hidden units

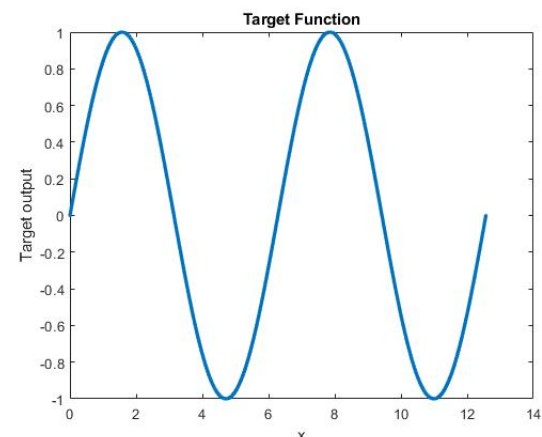

Figure 16: Sine Training Data

Figure 16 displays the plot of Input versus Target output, where $x$ represents the input within the range of 0 to $4\pi$, and the Target output corresponds to the sine of the inputs. Observing this plot might lead to the assumption that sigmoid or Tanh activations could potentially yield better results compared to ReLU or leaky ReLU. In the subsequent section, we aim to train a MOLF algorithm utilizing each of the aforementioned activations. Likewise, we will employ the AdAct algorithm using the corresponding initial activations. We plan to utilize one and ten hidden units for the Basic MOLF algorithm. Based on our analysis, we anticipate that sigmoid and Tanh activations will perform better for the sinusoidal data compared to ReLU and leaky ReLU. Therefore, we will commence with ReLU and leaky ReLU activations, followed by presenting the results obtained from sigmoid and Tanh activations.

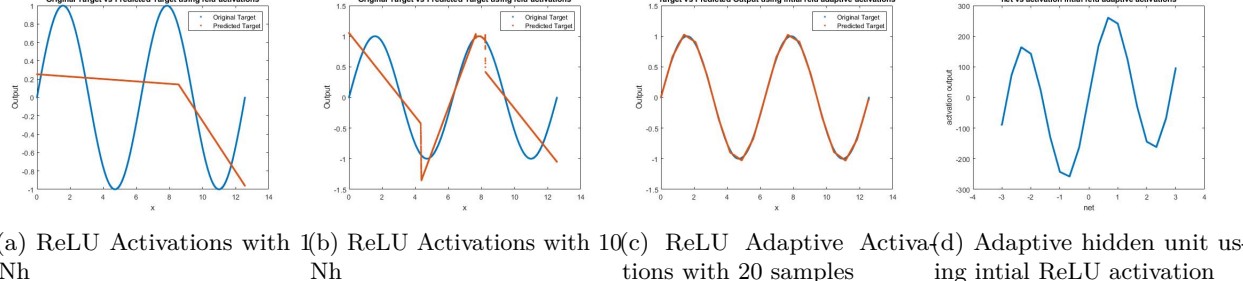

(a) ReLU Activations with 1 Nh    (b) ReLU Activations with 10 Nh    (c) ReLU Adaptive Activations with 20 samples    (d) Adaptive hidden unit using intial ReLU activation

Figure 17: Relu activations output

**ReLU Activation**  Figure 17 presents the results of the MOLF and AdAct algorithms using ReLU as the activation function. In 17a, the plot of input versus output using one fixed ReLU activation reveals that a single hidden unit with ReLU activation fails to approximate the simple sinusoidal function accurately. Increasing the hidden unit size to 10, as depicted in 19b, still shows ReLU activation struggling to approximate the sinusoidal curve. This limitation may arise from the fixed nature of the ReLU activation. Subsequently, we explore the use of ReLU as an initial activation function for adaptive activation. Through multiple experiments, we discovered that achieving better results for sinusoidal functions requires using a larger number of samples. In this particular case, we employed 20 samples. Figure 17c demonstrates that using adaptive activation improved the approximation compared to fixed ReLU activations. However, adaptive activations still struggle to accurately approximate the curve, which could potentially be addressed by utilizing more samples. Our experiments revealed that using 40 samples yields an approximation similar to fixed sigmoid activations, albeit with increased computational cost. Nonetheless, it's evident that adaptive activation with ReLU as the initial activation performs better than fixed activation. Moving on to the leaky ReLU activation with an alpha value of 0.01, since it resembles ReLU except for handling negative net values differently, we expect similar results. Furthermore, in Figure 17d, the single hidden unit after training showcases the net versus activation output resembling the output curve, but with notably higher activation outputs.

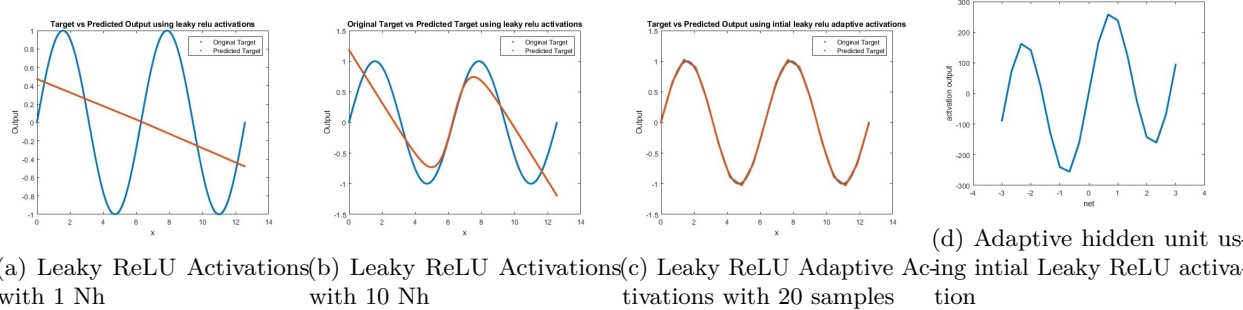

(a) Leaky ReLU Activations with 1 Nh
(b) Leaky ReLU Activations with 10 Nh
(c) Leaky ReLU Adaptive Activations with 20 samples
(d) Adaptive hidden unit using intial Leaky ReLU activation

Figure 18: Leaky Relu activations output

**Leaky ReLU Activation**  Figure 18 shows the MOLF and AdAct results using Leaky ReLU as the activation function where 17a is the input versus ouput using one fixed Leaky ReLU activation. We can observe from figure 18a and 18b leaky ReLU does not approximate the sinusoidal as similar to ReLU activation. But from figure 18c we can observe that adaptive activations work better again and almost similar to ReLU activations. Even the net vs activation output from figure 18d is similar to the ReLU activations.

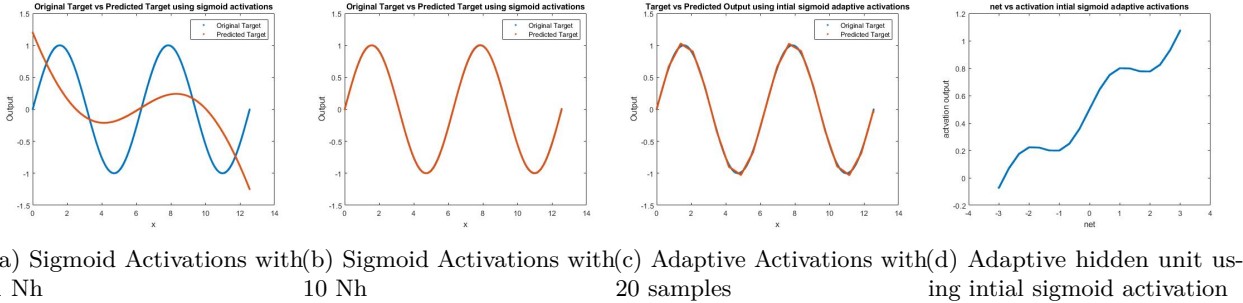

(a) Sigmoid Activations with 1 Nh
(b) Sigmoid Activations with 10 Nh
(c) Adaptive Activations with 20 samples
(d) Adaptive hidden unit using intial sigmoid activation

Figure 19: Sigmoid activations output

**Sigmoid Activation**  Figure 19 shows the MOLF and AdAct results where 19a is the input versus ouput using one fixed sigmoidal activation. We can observe that using one hidden unit is not sufficient to approximate the simple sinusoidal function. Now as we increase our hidden unit to 10 the training approximates a sinusoidal curve as it is evident from figure 19b. Now we will discuss the results with adaptive activations where the initial hidden unit is sigmoid. For the number of samples we chose 20 samples as we observed that if the

input output is a curve function, more number of samples are needed as our adaptive activation is piecewise linear. From figure 19c we can observe that the linear part of the sinusoidal curve is approximated well but because of the linearity of the adaptive activations it fails to approximate the curve accurately. This problem can be elimiated by adding more samples. From our experimentation, we observed that by using 40 samples the approximation is similar to the fixed sigmoid activations, but this comes with the increase computational cost. Figure 19d which shows the net vs activation graph which does not mimic exact similar graph to sigmoid but is close to sigmoid with smaller peaks at each of the positive and negative activation axis. Also thing to note is that the activation output values are very small as compared to the ReLU and leaky ReLU activations shown in paragraph C.1 and C.1

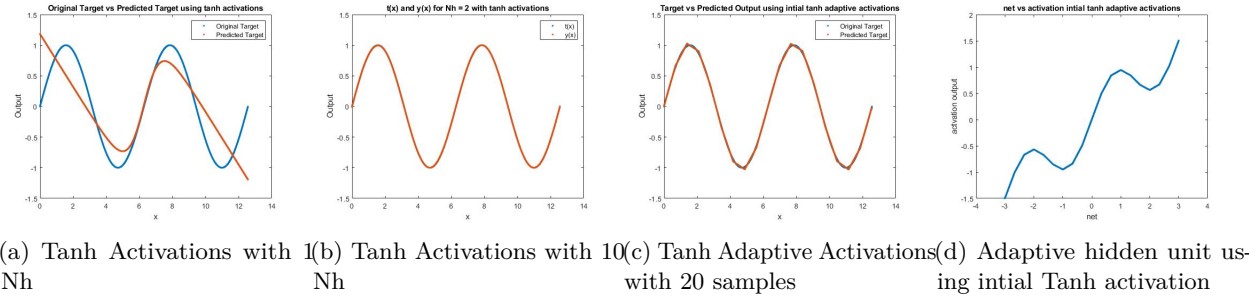

(a) Tanh Activations with 1 Nh

(b) Tanh Activations with 10 Nh

(c) Tanh Adaptive Activations with 20 samples

(d) Adaptive hidden unit using intial Tanh activation

Figure 20: Tanh activations output

**TanH Activation**  Figure 20 shows the MOLF and AdAct results where 20a is the input versus ouput using one fixed Tanh activation. We can observe that using one hidden unit is not sufficient to approximate the simple sinusoidal function. Now as we increase our hidden unit to 10 the training approximates a sinusoidal curve as it is evident from figure 20b. Now we will discuss the results with adaptive activations where the initial hidden unit is sigmoid. For the number of samples we chose 20 samples as we observed that if the input output is a curve function, more number of samples are needed as our adaptive activation is piecewise linear. From figure 20c we can observe that the linear part of the sinusoidal curve is approximated well but because of the linearity of the adaptive activations it fails to approximate the curve accurately. This problem can be elimiated by adding more samples. From our experimentation, we observed that by using 40 samples the approximation is similar to the fixed sigmoid activations, but this comes with the increase computational cost. Figure 20d which shows the net vs activation graph which does not mimic exact similar graph to sigmoid but is more closer to the sinusoidal curve than sigmoidal adaptive activation. Also thing to note is that the activation output values are very small as compared to the ReLU and leaky ReLU activations shown in paragraph C.1 and C.1

## C.2 Rosenbrock Approximation

To illustrate the approximation of the Rosenbrock function (7), we generate inputs with constant values of $a = 1$ and $b = 100$, creating 1000 uniformly distributed random samples for each of the two inputs. The output is determined by applying the Rosenbrock function to these inputs. For ease of training, we normalize both the inputs and outputs by centering them around a zero mean and standardizing them to a deviation of 1. We found that normalization didn't alter the obtained results significantly.

**Relu Activations**  Figure 21 displays the Rosenbrock approximation utilizing ReLU activations, while Figure 22 showcases the approximation using adaptive ReLU activations. In Subfigure 21a, a scatter plot illustrates the predicted versus actual output for the model trained with fixed ReLU activations. Meanwhile, Subfigure 22a presents results for the adaptive model trained using ReLU as initial activations with 11 samples. Comparing these scatter plots reveals that the adaptive ReLU model delivers superior approximation results compared to the model trained with fixed ReLU activation. Similarly, Subfigure 21b portrays a scatter plot for the predicted versus actual output obtained by the model using fixed ReLU activations for the second input. Concurrently, Subfigure 22b showcases results from the adaptive model trained using ReLU as initial

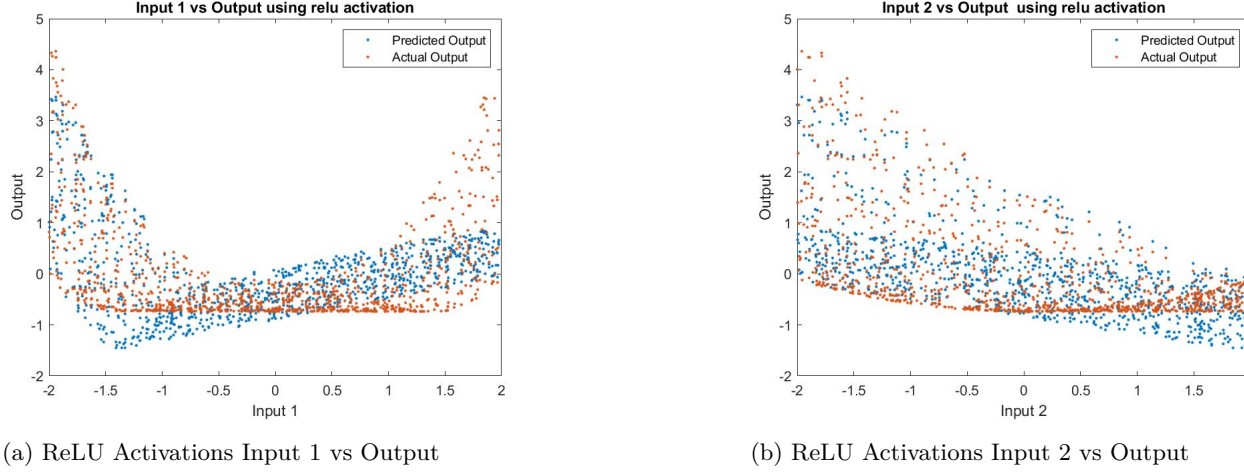

(a) ReLU Activations Input 1 vs Output

(b) ReLU Activations Input 2 vs Output

Figure 21: Rosenbrock Approximation with Fixed ReLU Activations

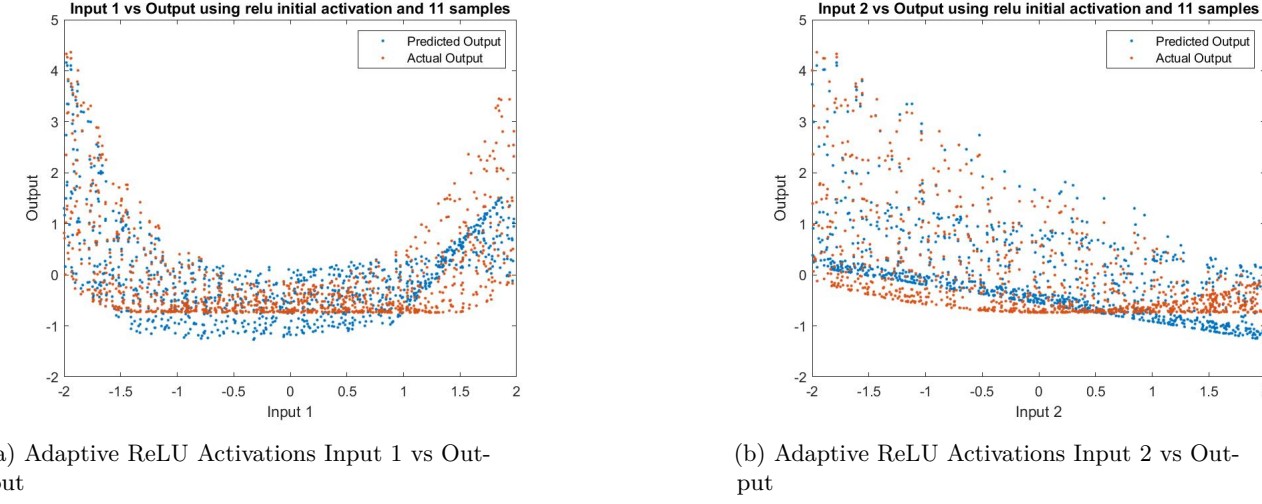

(a) Adaptive ReLU Activations Input 1 vs Output

(b) Adaptive ReLU Activations Input 2 vs Output

Figure 22: Rosenbrock Approximation with Adaptive ReLU Activations with 11 samples

activations with 11 samples. Once again, comparing these scatter plots reaffirms that the adaptive ReLU model outperforms the model trained with fixed ReLU activation in terms of approximation quality.

**Leaky Relu Activations** Figure 23 illustrates the Rosenbrock approximation employing Leaky ReLU activations, while Figure 24 demonstrates the approximation using adaptive Leaky ReLU activations. Subfigure 23a displays a scatter plot showcasing the predicted versus actual output from the model trained with fixed Leaky ReLU activations. In contrast, Subfigure 24a presents results for the adaptive model trained using Leaky ReLU as initial activations with 11 samples. Comparing these scatter plots reveals the superior approximation achieved by the adaptive Leaky ReLU model compared to the one trained with fixed Leaky ReLU activation. Likewise, Subfigure 23b features a scatter plot depicting the predicted versus actual output obtained by the model utilizing fixed Leaky ReLU activations for the second input. Meanwhile, Subfigure 24b showcases results from the adaptive model trained using Leaky ReLU as initial activations with 11 samples. Once again, the comparison between these scatter plots emphasizes the superior approximation performance achieved by the adaptive Leaky ReLU model over the model trained with fixed Leaky ReLU activation.

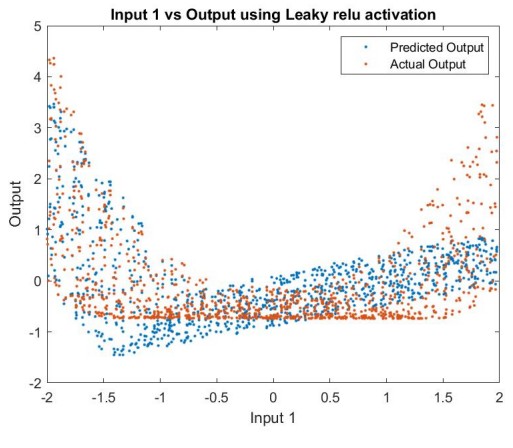 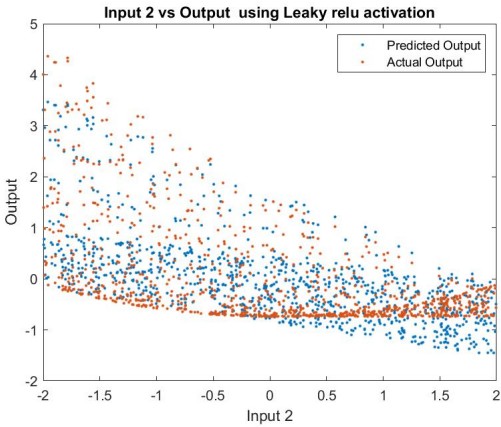

(a) Leaky ReLU Activations Input 1 vs Output  (b) Leaky ReLU Activations Input 2 vs Output

Figure 23: Rosenbrock Approximation with Fixed Leaky ReLU Activations

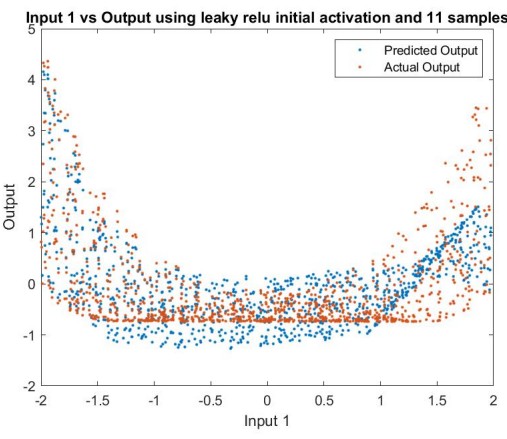 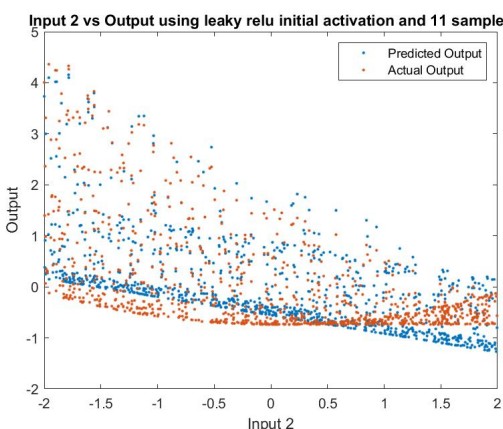

(a) Adaptive Leaky ReLU Activations Input 1 vs Output

(b) Adaptive Leaky ReLU Activations Input 2 vs Output

Figure 24: Rosenbrock Approximation with Adaptive Leaky ReLU Activations with 11 samples

**Sigmoid Activations** Figure 25 shows the Rosenbrock approximation using Sigmoid activations, and Figure 26 shows the Rosenbrock approximation using adaptive Sigmoid activations. Here, Subfigure 25a, shows a scatter plot of predicted versus the actual Output for trained model using fixed Sigmoid activations. Subfigure 26a shows results for the adaptive model trained using Sigmoid as initial activations using 11 samples. Comparing both the scatter plots, we can observe that the approximation results of the adaptive Sigmoid model is better than the model trained using fixed Sigmoid activation. Similarly, Subfigure 25b , shows a scatter plot of predicted versus the actual Output for the trained model using fixed Sigmoid activations for the second input. Subfigure 26b shows results for the adaptive model trained using Sigmoid as initial activations using 11 samples. Again, we can observe by comparing both scatter plots that the approximation results of the adaptive Sigmoid model is better than the model trained using fixed Sigmoid activation.

**Tanh Activations** Figure 27 illustrates the Rosenbrock approximation using Tanh activations, while Figure 28 showcases the approximation using adaptive Tanh activations. Subfigure 27a presents a scatter plot depicting the predicted versus actual output obtained from the model trained with fixed Tanh activations. Conversely, Subfigure 28a displays results from the adaptive model trained using Tanh as initial activations

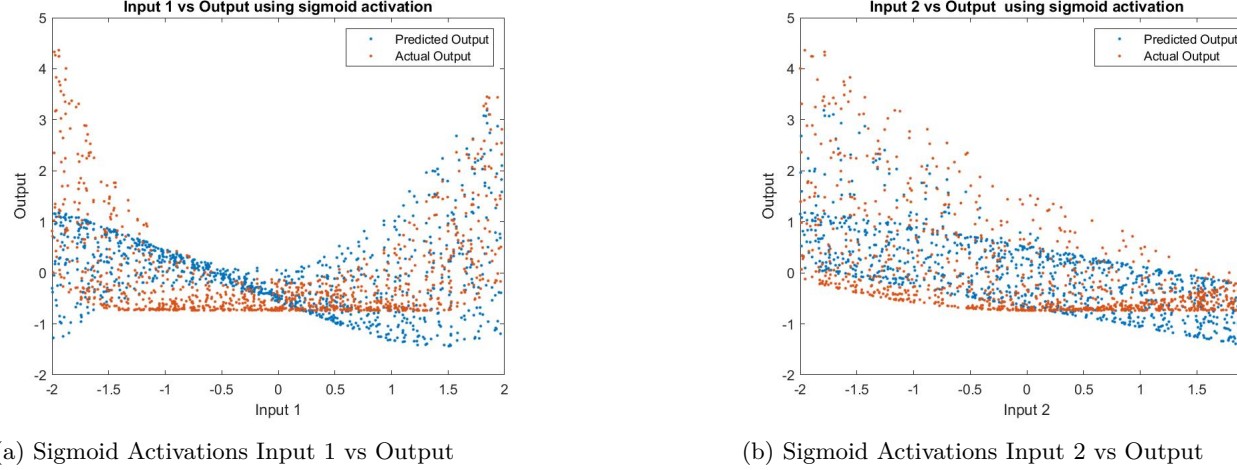

(a) Sigmoid Activations Input 1 vs Output

(b) Sigmoid Activations Input 2 vs Output

Figure 25: Rosenbrock Approximation with Fixed Sigmoid Activations

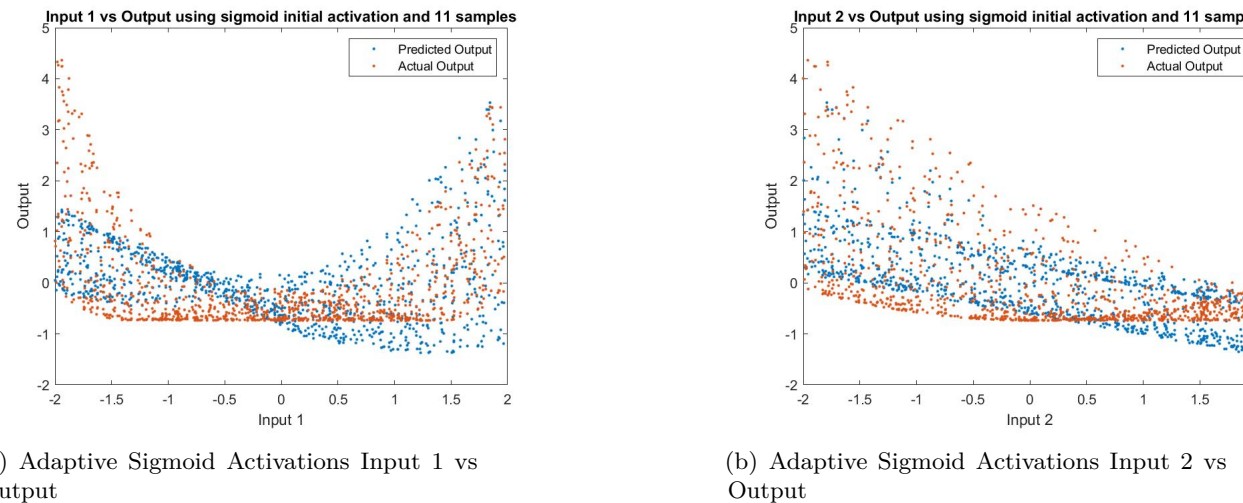

(a) Adaptive Sigmoid Activations Input 1 vs Output

(b) Adaptive Sigmoid Activations Input 2 vs Output

Figure 26: Rosenbrock Approximation with Adaptive Sigmoid Activations with 11 samples

with 11 samples. Comparing these scatter plots highlights the superior approximation achieved by the adaptive Tanh model compared to the one trained with fixed Tanh activation. Similarly, Subfigure 27b portrays a scatter plot showing the predicted versus actual output obtained by the model utilizing fixed Tanh activations for the second input. In contrast, Subfigure 28b exhibits results from the adaptive model trained using Tanh as initial activations with 11 samples. Once again, the comparison between these scatter plots emphasizes the superior approximation performance achieved by the adaptive Tanh model over the model trained with fixed Tanh activation.

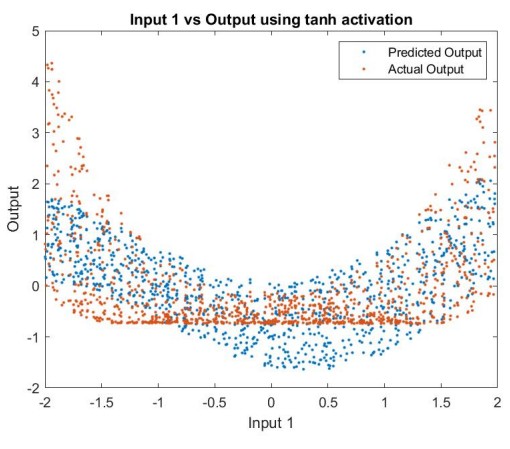
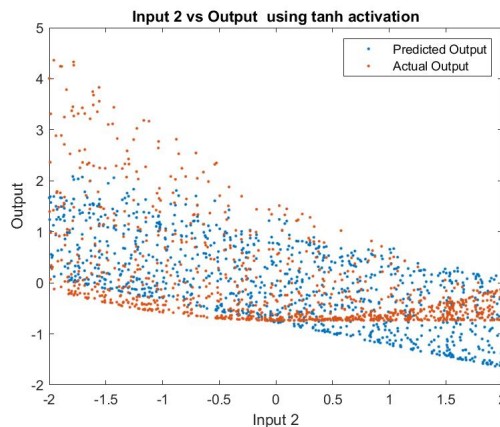

(a) Tanh Activations Input 1 vs Output

(b) Tanh Activations Input 2 vs Output

Figure 27: Rosenbrock Approximation with Fixed Tanh Activations

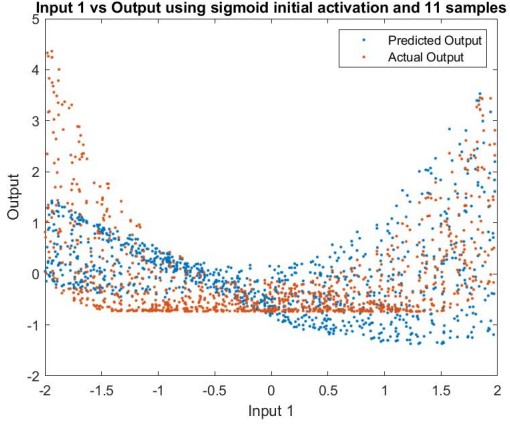

(a) Adaptive Tanh Activations Input 1 vs Output

(b) Adaptive Tanh Activations Input 2 vs Output

Figure 28: Rosenbrock Approximation with Adaptive Tanh Activations with 11 samples

