# OpenReview forum: "Optimizing Performance of Feedforward and Convolutional Neural Networks through Dynamic Activation Functions"
_TMLR — Rejected by TMLR_

### Review · Reviewer_yfM9 · 2023-10-22

**Summary Of Contributions:**

This paper studies the impact of different activation functions on the final output under the same neural network structure, and focuses on the study of piece-wise ​​linear activation functions. The authors conducted some relevant experiments, and some experimental results show that the piece-wise linear activation function performs slightly better.

**Audience:**

Yes

**Broader Impact Concerns:**

I have no concerns on the broader impact of this paper.

**Claims And Evidence:**

Yes

**Requested Changes:**

It will be great if the authors can make some discussion on the questions I listed above.

**Strengths And Weaknesses:**

I find the result interesting. I still have a few questions about the experiment in the paper.

1.I am not sure whether I understand the setting correctly. In the experiment of sine function, the networks only have 1 hidden unit and 10 hidden units?

2.I do not quite understand the difference between two subfigures from Figure15 to Figure21. Can the difference be measured by MSE?

3.Still in the experiment of Rosenbrock Approximation, is the model trained by 11 samples? If so, why not use more samples?

4.In the above two expeiments, what would happen if we use more units and more traning sample?

5.There are so many "??" in the paper.

---

> ### Author Response · Authors · 2023-12-08
> **Response-1**
>
> Firstly, thank you for taking out time to review the paper and provide your valuable comments. The paper  has incorporated all the suggested changes and we have uploaded the revised manuscript.
>
> 1.I am not sure whether I understand the setting correctly. In the experiment of sine function, the networks only have 1 hidden unit and 10 hidden units?
>
> Yes, we have experimented with various hidden units and reported only those two to give a spectrum of performance with lowest hidden units (1) upto a much higher number.
>
> 2.I do not quite understand the difference between two subfigures from Figure15 to Figure21. Can the difference be measured by MSE?
> This is a typo and we have fixed it in the updated document.
>
> 3.Still in the experiment of Rosenbrock Approximation, is the model trained by 11 samples? If so, why not use more samples?
> We experimented with multiple samples and  11 is the best one.
>
> 4.In the above two experiments, what would happen if we use more units and more traning sample?
> There is a upper threshold that we reach as we increase the number of hidden units and the training samples.
>
> 5.There are so many "??" in the paper.
> We have now updated the document with any missing references that was causing those ?? in the paper.
>
> Once again, thank you giving your comments on the paper. We agree that the manuscript was hard to read but the updated manuscript is a major revision from the previous one.

---

### Review · Reviewer_hsnt · 2023-11-15

**Summary Of Contributions:**

Apparently, the authors propose a new way to implement trainable activation functions for neural networks, based on a novel parametrization of piecewise linear functions. The method is demonstrated on toy functions, classification datasets and CIFAR-10, with various architectures, suggesting good performance.

**Audience:**

Yes

**Broader Impact Concerns:**

I do not see any broader impact concerns.

**Claims And Evidence:**

No

**Requested Changes:**

This is a sample of the reasons why the paper cannot be published in its current state, in my opinion:

Throughout, citations have their parenthesis in the wrong place, after the author's name rather than before, resulting in passages such as "deep recurrent neural networks(RNNs)Rumelhart et al. (1986a)Jordan (1997)Hopfield (1982)" in the introduction. This suggests that the authors didn't actually read the PDF before submitting it.

p. 5: What are the "direction vectors" P? What is the "gradient energy"? Also: what are the weight vectors W, Woh and Woi exactly? The previous section mentioned convolutional networks, but this doesn't look like a convolutional architecture?

p. 6: Please spell out the definition of the Hessian matrix.

Equation 26: Please re-state what p, n_p and k mean.

p. 8: figure 4 (should have a capital f in "figure)

Figure 5: Equation 27 suggests there should be one separate a for each hinge. Why is only one a and only one b shown in Figure 5?

p. 9 "more significant"  should be replaced with "higher" or "larger".

Section 4.2: Fix the multiple "??".

Section 4.2 is evidently the core of the paper but is incomprehensible and probably incorrectly typed. What is the "net function"? Is it the input to the activation function?  What is the "activation value" of a hinge? (From the rest of the text I vaguely sense that it is the value of the target output function, which is to be approximated, at the location of each hinge - if so, please state it).  We are told that "ns(1,k) [is] denoted as m1" - but in Equation 32 we see the term "ns(m1, k)"! Does it imply some recursion, or is it a typo? What, exactly, is ns, m1 and k?

We are told that "the calculation fails when the distance between the heights of two hinges is very close". Why is that? If two successive hinges have similar height, it simply means that the slope between these hinges is close to zero. Why would that be a problem? Why does the new proposed parametrization fix this problem?

Please explain in words what the complex equation 34 is doing, intuitively.

Section 4.4 evidently explains how to compute gradients. However, it is also corrupted and incomprehensible.  "as described by the search algorithm": Which search algorithm? "A search algorithm is used to find the correct m sample for a particular pattern’s hidden
unit is found": this is non-grammatical and incomprehensible. Again, which search algorithm is used?

Section 4.5's initial sentence (or double-sentence) is also ungrammatical and incomprehensible.

4.6.1: "We use 1 hidden unit for each training algorithm with 20 samples for the Adapt-ACT-OLF algorithm". What are these 20 "samples"? Earlier in the section "samples" seems to design training and testing points. Do we use separate training points for this algorithm?

Figure 12b suggests that a ReLU network with 10 hidden units can only produce strictly linear outputs. This is contrary to the basic theory of multi-layer perceptrons! With multiple hidden neurons, the output neuron should be able to produce at least some crude, piecewise-linear approximation to the target function (One example: one hidden neuron h1 produces ReLU(x-1), another hidden neuron h2 produces ReLU(x-2), and a third hidden neuron h3 produces ReLU(x-4), then the output y is simply h1 - 2*h2 + 2*h3; this output is crudely oscillating). Please explain exactly what the architecture is and why the output for the ReLU network is strictly linear.

Section 5: "Adapt-act-olf" is mentioned twice. Alse: what are those datasets? Where do they come from?

Again, what is "samples/Nh" for the Adapt-OLF method?

"In Section 5 we are told that "the computational cost is measured", but this doesn't seem to appear anywhere in the text? I did not see any description of the actual computational cost. This is particularly important because the new proposed method seems to rely on an additional search process (the mysterious "search algorithm" mentioned above) in the gradient calculation. Thus it is necessary to provide the respective computation times for the various algorithms. Where is this displayed in the paper?

**Strengths And Weaknesses:**

Strengths:

- The method seems novel (from what I understand)

- It may or may not provide an advantage over existing methods for trainable activation functions

Weaknesses:

- The paper is unfinished and incomprehensible. It is extremely surprising that such a paper was even sent for review, given the obvious unfinished state. Gibberish sentences are common, multiple terms in the equations are left unexplained, and some equations are almost certainly wrong (see below). As a result, it is impossible to understand, much less reproduce, the method.

- In addition to the paper's incomprehensibility, some crucial information (namely, the computational cost of the method) seems to be missing - at least I haven't found it anywhere.

---

> ### Author Response · Authors · 2023-12-08
> **Response-1**
>
> a) This is a sample of the reasons why the paper cannot be published in its current state, in my opinion.
>
> We agree with the reviewer on the incompleteness of the paper. Thank you taking time to review the paper. We have rewritten all the parts of the paper and in our opinion it’s in a much better shape especially after incorporating all the suggested changes.  We have uploaded the revised manuscript.
>
> b) Throughout, citations have their parenthesis in the wrong place.
> Thank you for comments. We have fixed the references for the entire paper.
>
> c) p. 5: What are the "direction vectors" P? What is the "gradient energy"? Also: what are the weight vectors W, Woh and Woi exactly?
> Thank you for the suggestion. We have added the reference (Martin Fodslette Møller. A scaled conjugate gradient algorithm for fast supervised learning. Neural networks , 6(4):525–533, 1993) and moved this to the appendix section.
> d) p. 6: Please spell out the definition of the Hessian matrix.
> We have added a reference in the updated manuscript.
>
> e) Equation 26: Please re-state what p, n_p and k mean.
> We have it now explained in page 5 and updated it in rest of the manuscript.
> f) p. 8: figure 4 (should have a capital f in "figure)
> Thank you for the comment. We have updated it in the document and verified this mistake in rest of the manuscript.
>
> g) Figure 5: Equation 27 suggests there should be one separate a for each hinge. Why is only one a and only one b shown in Figure 5?
> Thank you for the comment. We have added an explanation for this in the updated document.
>
> h) p. 9 "more significant" should be replaced with "higher" or "larger".
> This is fixed in the updated document.
>
> i) Section 4.2: Fix the multiple "??".
> Thank you for this comment. All the  ?? have been fixed in the updated document.
>
> j) Section 4.2 is evidently the core of the paper but is incomprehensible and probably incorrectly typed. Thank you for the comment. There was a typo in the draft and we have now fixed it. We have changed the word “sampled” to “hinges” and added an explanation in the document as follows.
> ‘The activations output corresponding to the net values between the first two hinges are calculated with $ns(1,k)$ denoted as $m_1$ and $ns(2,k)$ denoted as $m_2$. Similarly, the activations output between the next two hinges are calculated by denoting $ns(2,k)$ denoted as $m_1$ and $ns(3,k)$ denoted as $m_2$. We do this for $H$ hinges.
>
> k) We are told that "the calculation fails when the distance between the heights of two hinges...
> Thank you for the comment, we have added an explanation to this in the updated document.
>
> l) Please explain in words what the complex equation 34 is doing, intuitively.
> Thank you for the comment, we have added an explanation to this in the updated document.
>
> m) Section 4.4 evidently explains how to compute gradients.
> The search algorithm is now added in the appendix and reference is added.
>
> n) Section 4.5's initial sentence (or double-sentence) is also ungrammatical and incomprehensible.
> Thank you for this comment, we have now fixed it in the updated document.
>
> o) 4.6.1: "We use 1 hidden unit for each training algorithm with 20 samples for the Adapt-ACT-OLF algorithm". What are these 20 "samples"?
> This is an important point that we overlooked. We have now changed the word “samples” to “hinges” to avoid confusion in the updated document.
>
> p) Figure 12b suggests that a ReLU network with 10 hidden units can only produce strictly linear outputs.
> Thank you for the comment. Yes you are correct, we have mistakenly put the wrong graph. In the updated document, we have added the correct plot in the appendix.
>
> q) Section 5: "Adapt-act-olf" is mentioned twice. Alse: what are those datasets? Where do they come from?
> We have added the references for each of the datasets.
>
> r) Again, what is "samples/Nh" for the Adapt-OLF method?
> samples/Nh is a notation we have used to explain the number of hidden units (Nh) used
>
> s) "In Section 5 we are told that "the computational cost is measured", but this doesn't seem to appear anywhere in the text?
> The computational cost of adaptive activation is the total number of trainable parameters plus hidden units times number of hinges. For example - if we have a 10000 parameter model and out of which we choose 50 hidden units with adaptive activations and  10 hinges each. The total number of parameters would be 10000 + (50 * 10), which is equal to 10500 parameter model

---

### Review · Reviewer_qRGP · 2023-12-01

**Summary Of Contributions:**

The authors propose to utilize complex piece-wise linear activation functions to improve neural network performance.

**Audience:**

Yes

**Broader Impact Concerns:**

N.A.

**Claims And Evidence:**

No

**Requested Changes:**

The writing of the paper should be significantly improved before it is ready for publication. There are multiple "??" caused by missing reference links. The citation should also be fixed. I suggest adding "~" before each "\cite".
Additional suggestions:
Section 2-3.4 should be significantly reduced.
P7. Generally, we don't define PWL functions as being composed of ReLU functions.
P8. 1st line. Why "we see that PLU combines ReLU and Tanh" here?
P8,L6, what did you mean by "fixed PLU"? L9 Why "three linear segments". L10: what do you mean by "there is minimal or no training."
Section 3.6 L3 why "can equal"
Please define "sample points."
Fig 5. Shouldn't there be more than one set of parameters for this curve?
P9: 1st line, I assume No hinge is needed to approximate a linear output.
In the CIFAR experiment, it appears the ReLU models are deprived of the Softmax function that we generally utilize to show the advantage of PWL function-based models. I don't believe this approach is correct.

**Strengths And Weaknesses:**

Strength:
The empirical verification shows some advantages of PWL over regular activation functions.

Weakness:
The paper is tough to follow, with a lot of detail not adequately described. The advantage shown in the experiments likely came from unfair conditions given to the PWL-based models.

---

> ### Author Response · Authors · 2023-12-08
> **Response-1**
>
> We acknowledge the reviewer's comments regarding the paper. Thank you for dedicating time to reviewing our manuscript. We've extensively reworked all sections of the paper, and we believe it's vastly improved, especially after integrating all suggested changes. The revised manuscript has been uploaded.

---

> ### Author Response · Authors · 2023-12-16
> **missing section added**
>
> We found out that sub-section 4.4.1 (Shallow CNN results) under section 4.4 ( CNN Results) was missing. It was a mistake while compiling the latex file.   This subsection has now been added and updated in the revised document.
>
> We apologize for any inconvenience to the reviewers and action editor.

---

### Author Response · Authors · 2024-01-14
**significant delay in the paper**

To the Editor in Chief and Action Editor,

The paper reviews were posted over a month ago, and the authors have also done a detailed rebuttal and a revised manuscript.  We sincerely request the Action editor to let us know when we can expect a decision on this manuscript.  The manuscript has been delayed beyond the expected timeline.

---

### Decision · Action_Editor_EjvJ · 2024-01-07

**Recommendation:** Reject

**Comment:**

This paper investigates piecewise linear activation functions (PLA) for deep neural networks. The authors propose an adaptive activation algorithm (AdAct) in which slope values and point locations are learned during the neural network's training.
Initially, the paper received negative reviews, with major concerns raised about its presentation, unfinished state, undefined terms, broken references, and missing sections (e.g., section 4.4.1). Reviewers also pointed out limited experimental validation, required clarifications in the evaluation setup, and concerns about the computational cost of the approach. During the discussion period, the authors addressed some presentation issues and submitted an updated paper. However, reviewers remained dissatisfied, noting unconvincing computational cost evaluation and regretting the experimental validation limited to very small networks on problems. After the discussion period, there was a consensus among reviewers to reject the paper.

The AE has carefully reviewed the submission and discussions. The AE believes that the paper needs further polishing and proofreading. Despite improvements after the first round of review, the presentation can still be significantly enhanced. For example, PLA is used several times and is only introduced on page 3. The AE also suggests consolidating the validations of the claims, including a clear evaluation of the computational cost of the approach and validation on larger networks and datasets. Therefore, the AE recommends rejection.

**Audience:**

The submission addresses the design of learnable activation functions in deep neural networks, which is of interest for the TMLR audience.

**Claims And Evidence:**

The claims are not enough supported by evidence.

**Resubmission Of Major Revision:**

The authors may consider submitting a major revision at a later time.